# Retinoic Acid Receptors and the Control of Positional Information in the Regenerating Axolotl Limb

**DOI:** 10.3390/cells10092174

**Published:** 2021-08-24

**Authors:** Trey Polvadore, Malcolm Maden

**Affiliations:** Department of Biology & UF Genetics Institute, University of Florida, Gainesville, FL 32610, USA; polvadore@ufl.edu

**Keywords:** retinoic acid, retinoic acid receptor, RARα, positional information, axolotl, limb regeneration

## Abstract

We know little about the control of positional information (PI) during axolotl limb regeneration, which ensures that the limb regenerates exactly what was amputated, and the work reported here investigates this phenomenon. Retinoic acid administration changes the PI in a proximal direction so that a complete limb can be regenerated from a hand. Rather than identifying all the genes altered by RA treatment of the limb, we have eliminated many off-target effects by using retinoic acid receptor selective agonists. We firstly identify the receptor involved in this respecification process as RARα and secondly, identify the genes involved by RNA sequencing of the RARα-treated blastemal mesenchyme. We find 1177 upregulated genes and 1403 downregulated genes, which could be identified using the axolotl genome. These include several genes known to be involved in retinoic acid metabolism and in patterning. Since positional information is thought to be a property of the cell surface of blastemal cells when we examine our dataset with an emphasis on this aspect, we find the top canonical pathway is integrin signaling. In the extracellular matrix compartment, we find a MMP and several collagens are upregulated; several cell membrane genes and secretory factors are also upregulated. This provides data for future testing of the function of these candidates in the control of PI during limb regeneration.

## 1. Introduction

The axolotl (*Ambystoma mexicanum*), a salamander native to Mexico, is capable of extraordinary feats of regeneration including complete regeneration of severed limbs. Regardless of where the amputation occurs along the proximal–distal (PD; shoulder to hand) limb axis, only the missing tissue is regrown. This indicates that cells in the limb blastema, a conical assembly of proliferating cells that will form the regenerated tissue, are somehow aware of their position along the PD axis. This property is reflected in the rule of distal transformation, which states that regeneration can only occur in the proximal-to-distal direction, as blastema cells can only create cells with an identity more distal than their own. An amputation between the elbow and wrist, for example, is unable to regenerate a second elbow as those tissues would require more proximal identities. While this phenomenon has been widely studied, the molecular mechanisms that govern the establishment, maintenance, and interpretation of positional information (PI) are still poorly understood.

Unlike development, where concentration gradients of extracellular morphogens are considered to be the means by which positional information is encoded, in regeneration, the cell surface of connective tissue blastemal cells has become the focus of attention. For example, when a wrist blastema is grafted proximally to the upper arm level and the limb is then amputated through the upper arm, the grafted blastema will move distally as the limb regenerates and then cease moving and integrate at the wrist level from which it originated [1]. When a blastema is cut off from the limb, rotated 180 degrees around the circumferential axis, and placed back on the limb stump, it will frequently de-rotate and end up in the same position at which it started [2]. When proximal and distal blastemas are cultured in vitro, proximal blastemas engulf distal blastemas, while blastemas from similar positions simply fuse with each other [3], suggesting there is a gradient of cell adhesion along the PD axis, which is used to interpret positional information through cell surface interactions.

An exception to the rule of distal transformation was found when distal blastemas were treated with retinoic acid (RA), a derivative of vitamin A, and regenerated an entire arm composed of all three limb segments (upper arm, forearm, and hand). This respecification of distal identity to a proximal state by RA was originally seen in axolotl limbs and found to be dependent on both the concentration of RA and length of treatment (Figure 1) [4,5]. In an attempt to understand the molecular basis of respecification, a subtractive screen was performed between untreated and RA-treated distal blastemas in the newt (*Notophtalmus viridescens*). In accordance with the suggested location of PI, this screen identified *prod1*, a gene encoding a cell surface protein, which was upregulated approximately 15-fold after RA treatment [6]. Further experimentation showed that removing Prod1 from the membrane or treating cells with a blocking antibody changed the engulfment behavior of distal cells, and over-expression of *prod1* in the blastema cells transformed distal cells into more proximal cells [7]. However, the axolotl homolog to newt Prod1 is not membrane-bound and does not appear to have direct effects on PI [8], so the search for a positional identity gene in the axolotl continues. Recent studies have identified differential gene expression in the axolotl limb blastema following RA treatment [9], but it has proven difficult to parse positionally relevant genes from the many genes involved in RA signaling.

We sought to perform another differential expression analysis that could better subset positionally relevant genes from the overall response to RA treatment of the limb blastema. In the early 1990s, synthetic RA agonists were designed to specifically bind to only one of the three known retinoic acid receptors (RARs) when administered at low concentrations [10,11]. The three agonists we have used here are Am580 (also known as CD336), CD2019, and CD437, which selectively bind to RARα, RARβ, and RARγ, respectively. We treated axolotl forelimbs amputated mid-radius/ulna with each RAR agonist and compared their effects on limb duplication that follows PI respecification. Our results indicate that all three agonists can induce complete limb duplication at 250 nM treatments, but only activation of RARα by Am580 consistently results in complete PD duplication at 25 nM treatment.

Following treatment of axolotl blastemas with 25 nM Am580, and thereby minimizing off-target gene expression induced by RARβ and RARγ, we extracted total RNA from only the blastema mesenchyme where the PI is known to reside. Am580 treatment induced significant differential expression in 3637 genes by RNA-sequencing, of which 2580 could be matched to an available reference sequence. A subset of 505 annotated genes was analyzed and revealed several cell adhesion, cell membrane, matrix, and secretory proteins. We also identified 23 unknown genes with homology limited to the genus *Ambystoma* and differential expression of fourfold or greater, which may represent candidates for future studies.

## 2. Materials and Methods

### 2.1. Axolotl Care

#### 2.1.1. Axolotl Procurement and Husbandry

Albino and wild type axolotls were purchased from the Ambystoma Genetic Stock Center at the University of Kentucky at sizes ranging from hatchlings (1 cm) to large juveniles (10–13 cm). All axolotls were housed individually for the duration of the experiments to prevent cannibalistic limb loss. Each animal was kept in room temperature, 40% Holtfreter’s solution (HS). Axolotls were fed a mixture of live brine shrimp, frozen bloodworms, and fish pellets. All axolotl experimentation was performed in compliance with the University of Florida’s Institutional Animal Care and Use Committee (protocol number 201810351, approved 25 September 2018).

#### 2.1.2. Limb Amputations

Limb amputations were conducted with a single cut to the mid-zeugopod following systemic anesthetization in tricaine-s (Sigma-Aldrich, St Louis, MO, USA) dissolved in HS. The concentration of tricaine-s used was dependent on axolotl length and ranged from 300 to 1000 mg/L.

#### 2.1.3. Systemic RAR Agonist Treatment

Following limb amputation, the axolotls were placed in HS for 24 h to recover. Animals in the treatment groups were then placed in the working RAR agonist solution for 3–5 days post amputation and the animals in the control group were placed in HS with DMSO (volume equal to amount of agonist solution used in treatment animals). The solutions were changed every 24 to 48 h. All animals were then returned to HS until limb regeneration was complete or until tissue was harvested.

### 2.2. Solution Preparation

#### 2.2.1. Victoria Blue Stain

Victoria Blue stain was prepared by dissolving 1% *w*/*v* Victoria Blue dye in acid alcohol. 

#### 2.2.2. RAR Agonist Solutions

Stock solutions of all-*trans*-RA, Am580 (Sigma-Aldrich, St. Louis, MO, USA), CD271, CD2019, and CD437 (Tocris Bioscience, Bio-Techne Corp, Minneapolis, MN, USA) were prepared to 1 mM by dissolving in DMSO and stored at −20 °C. Working solutions were prepared by dilution with HS to achieve the desired treatment concentrations ranging from 10 nM to 250 nM.

### 2.3. Limb Analysis

#### 2.3.1. Limb Collection and Staining

Regenerated limbs were collected and fixed in 10% neutral buffered formalin (NBF) for at least 24 h at 4 °C. The limbs were bleached overnight at room temperature in 30% hydrogen peroxide. Each limb was dehydrated in 50% ethanol 1 h, treated with 1% HCl in 70% ethanol for 2 h, and then stained with Victoria blue for 45 min. After dehydrating to 100% ethanol, the tissue was cleared in methyl salicylate.

#### 2.3.2. Duplication Scoring

Following collection of the limbs and staining with Victoria blue, the regenerates were photographed and visually evaluated for proximodistal respecification. Duplication scores (Figure 1) were assigned ranging from 0 (normal regenerate with no duplication) to 5 (complete duplication from stylopod to autopod) [4]. Intermediate scores are assigned according to the degree of duplication.

### 2.4. RNA Sequencing 

#### 2.4.1. RNA Extraction

For the sequencing experiment, mesenchyme tissue was collected only from the right forelimb blastemas, which had the epidermis removed, of both Am580 and DMSO-treated axolotls. The left contralateral limb was allowed to continue regeneration and served as confirmation that respecification had taken place, since contralateral pairs of limbs from any one animal always behaved identically. The right mesenchymes were immediately placed in RNALater, incubated at 4 °C overnight, and then stored at −20 °C. Total RNA was extracted using a RNeasy Fibrous Tissue Mini Kit (QIAGEN 74704, Qiagen, Germantown, MD, USA) according to the manufacturer’s protocol. A total of 675–2889 ng of RNA was extracted from each sample (means of 1350 ng (*s* = 574 ng) and 2168 ng (*s* = 563 ng) for Am580 and DMSO-treated tissue, respectively), as quantified using a Qubit fluorometer. Agilent TapeStation analysis assigned RINs ranging from 9.5 to 10.0 (mean RIN of 9.9, *s* = 0.14). From each treatment group of ten axolotls, RNA extracts were used for RNA-sequence library construction (*n* = 6).

#### 2.4.2. Library Construction and RNA Sequencing

An Illumina TruSeq stranded library was constructed via poly-A selection for each individual blastema sample for a total of 12 libraries (six from the Am580-treated group and six from the DMSO-control group). The libraries were sequenced on the Illumina NovaSeq6000 platform to yield approximately 642 million paired-end, 150 bp reads (mean of 53.5 M read pairs (*s* = 6.5 M read pairs) per library). Library construction and sequencing were performed by HudsonAlpha Discovery, Huntsville, AL, USA. 

#### 2.4.3. Read Processing and Quality Control

All reads were evaluated with FastQC [12] for base sequence quality, adapter content, and duplication level to identify any library-level errors. Adapter sequences and low-quality bases were trimmed from each read using Trimmomatic [13] and then re-evaluated with FastQC. Surviving paired-end reads were not merged, and single-end reads were excluded from downstream analyses.

#### 2.4.4. Data Processing and Visualization

Data processing was performed in R [14] using RStudio [15]. Data visualization was accomplished with the R graphics package ggplot2 [16].

### 2.5. RNA-Seq Analysis

#### 2.5.1. Transcriptome Assembly

Reads in each fastq file were aligned to the AmexG_v3.0.0 axolotl genome [17] using HISAT2 [hisat2/2.1.0, -q -dta] [18]. The resulting SAM files were converted to BAM files using samtools [samtools/1.8, samtools view -S], sorted [samtools sort], and indexed [samtools index -c] [19]. Due to the large size of the individual chromosome arms, it was necessary to create a CSI index instead of the standard BAI index. 

A combined transcriptome for treated and control blastemas was constructed using StringTie [stringtie/1.3.4d] [20,21] and gffread [gffread/0.9.8c] [22]. The GenePred annotation file provided with the genome was downloaded (https://www.axolotl-omics.org/dl/AmexG_3.0.0_FinalGeneSet.gp) and used to generate new transcript names that retained the reference names when possible and utilized the “MSTRG” name from StringTie for novel transcripts. Custom R [14] scripts were used in RStudio [15] to manipulate the reference GTF and StringTie output.

#### 2.5.2. Differential Expression Analysis

Transcript counts were produced with featureCounts [subread/1.6.2, featureCounts -s 2 -p -t exon -g gene_id] [23] using the transcriptome annotation file generated by StringTie. Custom R scripts were used to clean the featureCounts output and differential expression analysis was conducted with edgeR [24]. Low expressed reads were removed from the analysis by filtering for transcripts that did not have at least ten counts in at least six samples. The differential expression model fit a quasi-likelihood method and significance was determined with the [edgeR/3.26.8, glmQLFit, adjust.method = “BH”, *p* value = 0.05]. A total of 3454 genes were identified as differentially expressed at a log2 fold change greater than zero, with 1650 genes upregulated and 1804 downregulated. An additional 18,693 genes were present but did not exhibit significant differential expression.

#### 2.5.3. Transcript Annotation

All differentially expressed transcripts identified by edgeR were annotated with BLASTx [ncbi_blast, blastx -db swissprot -outfmt 10 -max_target_seqs 1] against the SwissProt database, supplementing the existing genome annotation. The top hit for each gene was retained for downstream analyses.

#### 2.5.4. Pathway Analysis

Differentially expressed genes with human Entrez designations were loaded into Ingenuity Pathway Analysis (IPA; QIAGEN) and subjected to pathway analyses with various fold change cutoffs. Using only genes with at least a twofold change, only 275 analysis-ready molecules were available, whereas decreasing the foldchange cutoff to 1.5 increased the analyzed genes to 485. IPA was used primarily for the identification of significantly activated or inhibited canonical pathways.

### 2.6. Data 

The data are currently being submitted to NCBI and the accession numbers will be provided during review.

## 3. Results

### 3.1. Human and Axolotl RARs Are Homologous

As the agonists were designed to selectively bind to human RARs [10,11], we sought to determine if agonist specificity would likely be retained when paired with the axolotl RAR homologs. Three amino acid residues within the ligand-binding domain of the human RARs are responsible for the binding specificity of each agonist [25]. Sequence similarities between human and axolotl RARs are high (87%, 90%, and 83% for RARα, RARβ, and RARγ, respectively) and axolotl RARs contain the same three specificity-granting amino acid residues, indicating that the agonists show similar selectivity with axolotl RARs (Figure 2).

### 3.2. Effects of Selective RAR Agonists on Limb Regeneration

As previous studies have shown that individual RARs are responsible for the formation of specific phenotypes, we hypothesized that positional information can be proximalized by the activation of a single RAR [27,28,29]. Forelimb stumps, 24 h post-amputation, were treated with synthetic agonists Am580 (RARα agonist), CD2019 (RARβ agonist), CD271 (RARα and RARβ dual-agonist), CD437 (RARγ agonist), or all-*trans*-RA. Systemic treatment of 3–5 cm-long axolotls at a concentration of 250 nM agonist resulted in varying degrees of proximal–distal limb duplication following mid-zeugopod amputation of both forelimbs. Qualitative, visual scoring of Victoria blue-stained regenerates were used to compare the proximal–distal (PD) duplication of each treatment (Figure 3). The limbs treated with 250 nM RA received a mean score of 2.5 out of 5, while the control limbs treated with equal volumes of the solvent DMSO received a score of 0 with no indication of PD duplication. While all four agonist-treated groups had at least one limb with complete duplication, only Am580 (RARα) and CD2019 (RARβ)-treated groups were comprised of complete duplication in all treated limbs. The RARγ agonist, CD437, had the greatest variation with a mean score of 2.7, and the RARβγ dual-agonist CD271 resulted in a mean score of 3.8. These results demonstrate successful respecification of positional identity to a more proximal state in all treatment groups as they all contain duplicated proximal elements.

### 3.3. Systemic Treatment of 25 nM Am580 Consistently Respecifies Positional Information

While the 250 nM agonist treatments suggest that activation of both RARα and RARβ lead to the highest degree of PD respecification, the possibility remained that only a single receptor is truly responsible for proximalization. As previous studies [10] have shown that the specificity of each agonist for a single RAR is highly dependent on agonist concentration and that, at any one concentration, the agonists elicit comparable binding [11], we repeated the systemic treatments at 25 nM on slightly larger axolotls (6–8 cm). When the agonists are more selective at this lower concentration of 25 nM, only activation of RARα by Am580 consistently results in PD duplication with a mean score of 4.9, compared to the CD2019 and CD437 treatment groups with mean scores of 0.1 and 0.3, respectively (Figure 4A). Comparing the 250 nM (Figure 3) and 25 nM (Figure 4A) treatments reveals that the 10X dilution results in a decrease in limb duplication following CD2019 and CD437 treatments, while there is minimal difference between the duplication scores of Am580-treated limbs. Representative Victoria blue-stained limbs from each treatment group at both 250 nM and 25 nM are compared in Figure 4B.

In search of a minimum concentration that could elicit the PD duplication seen above, hatchling axolotls (1–2 cm long) were treated with 10 nM of each agonist. These results revealed that the effects of Am580 are vastly reduced at such a low concentration. Many limbs treated with Am580 developed a cartilaginous spike on the ulna but generally did not exhibit any sign of proximalization. Four limbs treated with CD437 had a clear gap in either the regenerated radius or ulna. These results indicate that systemic agonist treatment of hatchlings at 10 nM is insufficient to induce duplication of the PD limb axis.

Experimenting on hatchling axolotls is convenient due to their low cost and rapid limb regeneration, but their tiny blastemas present a problem for surgical manipulation and molecular extraction/analysis. In order to collect enough tissue for larger studies, such as RNA-sequencing as we performed here, it is more convenient to work on a smaller number of larger axolotls that will yield larger blastemas. To determine if 25 nM systemic treatments is enough to respecify large juvenile limbs, 16 axolotls of approximately 10 cm in length were treated with either Am580 or DMSO. Despite the low concentration of only 25 nM, the large size of the axolotl, and the systemic delivery method that required absorption through the gills or skin, every Am580-treated limb showed complete PD duplication. We conclude that treatment with 25 nM agonist is best able to consistently induce complete PD respecification while maintaining agonist specificity across a range of sizes of axolotls.

### 3.4. RNA-Sequencing of Blastema Mesenchyme Tissue Following Am580 Treatment

#### 3.4.1. Systemic Am580 Treatment

To better understand the role of RA in respecifying positional information, we employed the Am580 agonist that selectively binds and activates RARα. The primary goal of using a selective agonist instead of RA was to minimize activation of off-target genes that are not involved in specifying positional identity. Twenty large juvenile axolotls (10–13 cmlong) were selected for systemic Am580 or DMSO treatment and blastema mesenchyme tissue collection. After amputating both forelimbs and resting for 24 h, 10 axolotls were placed in 25 nM Am580 in HS, and the remaining 10 axolotls were placed in an equal volume of DMSO in HS (Figure 5A). After five days of treatment, the axolotls were returned to HS for 24 h. The right forelimb blastemas were then collected from all 20 axolotls, the epidermis was removed from the blastemas and the mesenchyme stored in RNALater. The left forelimbs were left intact to regenerate to serve as observational controls for the efficacy of respecification since contralateral limbs behave identically. At the completion of regeneration, the left forelimb regenerates were collected, stained with Victoria blue, and scored based on proximodistal duplication. Scores for 9 Am580-treated (1 axolotl died prior to limb collection) and 10 control limbs show an increase in duplication of Am580-treated axolotls compared to the DMSO control (Figure 5D).

#### 3.4.2. RNA-Sequencing Metrics

The six limbs from the Am580 treatment group that showed the greatest level of duplication (Figure 5B) as well as six limbs from the DMSO control group, all scoring 0 with no duplication, were selected for RNA extraction and RNA-sequencing. Approximately 642 million paired-end, 150 bp reads were obtained from 12 libraries (mean of 53.5 M read pairs per library), with each library constructed from total RNA collected from the mesenchyme tissue of an individual blastema. All 12 libraries passed quality control analysis with fastQC [12], showed improvement following quality and adapter trimming with Trimmomatic, and were retained for downstream analyses.

#### 3.4.3. Transcriptome Assembly and Differential Expression Analysis

Using the axolotl genome (version AmexG_v3.0.0) as a reference, a transcriptome was assembled from reads in both the Am580 and DMSO treatment groups. Trimmed, paired-end reads were aligned to the genome using HISAT2 and the transcripts assembled using StringTie. A transcriptome was created using gffRead and gffCompare. The final assembly is composed of 193,947 total transcripts, including 11,962 novel transcripts identified by StringTie.

From 36,917 genes identified by featureCounts, 3637 were determined by edgeR to be differentially expressed in the Am580-treated animals compared to the DMSO-control groups at an adjusted *p*-value of 0.05. Only 2580 were matched to a significantly similar homolog by BLASTx, indicating that many of the identified genes are either novel to the axolotl or simply not yet annotated in the SwissProt database. Of these 2580 differentially expressed genes, 1177 are upregulated and the remaining 1403 are downregulated. A restricted list of 505 genes was created by selecting all genes with at least a twofold change in expression, and the top thirty differentially expressed genes are presented in Table 1.

### 3.5. Differentially Expressed Genes

#### 3.5.1. Integrin Signaling

In a pathway analysis conducted using all significant differentially expressed genes from our Am580-treated blastemas, the top canonical pathway identified by IPA is integrin signaling. A total of 63 genes associated with integrin signaling are found among our differentially expressed genes, including 10 transmembrane receptors (Table 2). All but two integrin signaling receptor genes, *itgav* and *itgb3*, are upregulated following RARα activation. Integrins are heterodimeric glycoprotein complexes formed from an alpha and beta subunit, and they function as receptors on the cell surface to mediate cell migration, adhesion, and the construction of extracellular matrix (ECM). One of the most common integrin ligands is fibronectin, and we identified *Lrfn5*, the gene for a fibronectin type III protein, as downregulated 1.28-fold following Am580 treatment. The upregulation of nine integrin subunit genes and *Flrt2* (upregulated 0.51-fold), a fibronectin receptor gene, and the downregulation of the ligand *Lrfn5* suggest that integrin signaling may play an important role in the establishment of new, proximalized positional information following RA treatment.

#### 3.5.2. Extracellular Matrix Genes

Positional information in the axolotl limb is known to be located in the connective tissue mesenchyme, but recent evidence suggests that it may also be present in the extracellular matrix (ECM). A gain-of-function assay was conducted in vivo using the ALM technique and decellularized axolotl ECM [30]. It was found that grafting ECM with disparate positional information to an innervated wound could maintain blastema outgrowth and form rudimentary limb elements. Decellularized mouse ECM from corresponding limb locations was also able to induce blastema growth in the axolotl, indicating similar mechanisms of positional identity interpretation. To find avenues for further exploration of the roles that the ECM may play in PD positional identity, we identified several collagen and matrix metalloproteinase (MMP) genes that are differentially expressed following Am580 treatment. Three identified proteinases, *Mmp9*, *Mmp14*, and *Mmp28*, are downregulated 2.27, 0.64, and 0.83, respectively, and *Mmp21* is upregulated 8.68-fold. As previously discussed, *Mmp9* is upregulated by Prod1 treatment in urodeles, so its downregulation here may prove an interesting avenue for study. The ECM itself is made of mostly collagens, and we identified seven differentially expressed collagen components. Five components, *Col5a1*, *Col5a3*, *Col6a1*, *Col6a2*, and *Col6a6*, are upregulated 3.63-, 2.09-, 1.07-, 0.85-, and 1.98-fold, respectively. The remaining two components, *Col4a5* and *Col4a6*, are downregulated 1.11- and 1.38-fold, respectively.

#### 3.5.3. Cell Membrane and Secretory Protein Coding Genes

To identify candidate genes with direct roles in establishing, maintaining, and interpreting positional information, we searched for upregulated genes with at least a twofold change that encode secretory proteins or membrane-bound proteins with a domain exposed to extracellular space. Positional information is thought to be maintained on the cell surface of mesodermal, fibroblast-like connective tissue cells, such as Prod1 in the newt, and discovering a suitable membrane protein in the axolotl would be a substantial finding. While this assay did not reveal a single obvious gene candidate, 15 upregulated membrane proteins were found with known functions in cell signaling and another 7 with structural roles involving cell adhesion (Table 3). Proteins secreted into the extracellular space could be involved in cell–cell signaling that relay positional information to nearby cells, so a thorough search for genes that encode potentially secreted proteins was performed. We identified a total of 51 differentially expressed, secretory protein-coding genes, including 29 genes upregulated at least twofold (Table 4).

#### 3.5.4. Unknown Genes

Many differentially expressed genes could not be linked to a known protein sequence either from the genome annotation file or an additional BLASTx analysis. Twenty-three unannotated transcripts with at least a fourfold change following Am580 treatment were selected for BLASTn analysis to determine if these transcripts are derived from genes novel to the axolotl or if they are conserved in other species. It is possible that key genes involved in limb regeneration and positional information may be novel to the axolotl or the genus *Ambystoma*, and these genes will be overlooked if only annotated transcripts are analyzed. A standard BLASTn analysis returned no significant result for 14 transcripts, hits limited to species of the genus *Ambystoma* for 4 transcripts, and hits to only known axolotl sequences for the remaining 5 transcripts. While some BLASTn results included homologous matches to sequences in other organisms, the regions matching to the query were very short. All hits with a match smaller than 10 percent of the query length were ignored and not included in this analysis. These 23 genes cannot be identified through homology with known sequences and their function in limb regeneration and positional identity remain a complete mystery. Future work should consider investigating these genes as they undergo considerable differential expression during Am580 treatment.

## 4. Discussion

We have shown here that the effect of RA on respecifying the PD axis of the regenerating axolotl limb and thus duplicating the elements that are regenerated is transduced by only one of the three RARs, namely RARα. We arrived at this conclusion by decreasing the concentration of each of the RAR agonists until complete selectivity was obtained, at a concentration of 25 nM. At this concentration, the RARα agonist fully duplicated the regenerating limb such that after amputating through the middle of the lower limb, the regenerate contained all the elements of the limb, beginning from the proximal stylopod instead of just replacing the elements that were amputated. In contrast, the other two RAR agonists are completely inactive at this concentration and had no effect on the regenerating limb. At a 10-fold higher concentration, however, all three agonists were active at inducing duplications and this is due to the typical loss of selectivity of pharmacological agents which occurs at saturating conditions [11]. This explains why a previous use of one of the RAR agonists, a RARγ agonist, concluded that this was the receptor involved, but without comparing any of the other agonists nor performing a concentration experiment [9].

We have previously shown by microarrays in the neural stem cells of the mouse brain that these RAR agonists induce a different, although not unique, set of downstream gene targets [31]. For example, after treatment with a RARα agonist, a total of 3155 genes were responsive to the treatment, of which 1439 were specific to RARα, the remainder overlapping with RARβ and RARγ agonist treatment. So, while not generating a unique dataset, the use of this RARα agonist certainly reduces the gene targets compared to the use of the pan-agonist RA as has been used in previous experiments of this type [6,9]. Therefore, we proceeded to identify the gene targets of RARα during limb duplication and, in addition, as a further refinement, we used RNA isolated only from the blastemal mesenchyme rather than the whole blastema, since the mesenchyme is the tissue which is responsible for the effects of RA and not the epidermis [32].

We identified 2580 differentially expressed genes, less than half of which were upregulated and more than half were downregulated. The top upregulated gene was *Cyp26A*, closely followed by *Rarβ*, both of which are in the RA regulation pathway. Interestingly, in a microarray experiment [9] where blastemas were treated with RA, rather than the RARα agonist, a more extensive range of genes in the RA pathway were induced. For example, all three *Rar*s and *Crabp2* were induced in this work, but we only saw one receptor, *Rar*β, and neither of the *Crabp* genes, suggesting that we have indeed identified a sub-set of RA responsive genes.

In terms of known positional genes, *Meis1* and *Meis2* are upregulated following PD limb duplication initiated by RA treatment, and their overexpression proximalizes limb tissue [33]. In our Am580-treated samples, we identified *Meis1* and *Meis2* as significantly upregulated by 0.72- and 2.73-fold, respectively. We also identified *Pbx1*, a transcription factor that dimerizes with Meis proteins, as upregulated 1.83-fold. Fgf8 plays a role in keeping limb bud cells in a distal state as it represses *Meis* expression, represses distal Hox gene expression and is itself repressed by RA. The top downregulated gene in our RARα analysis was *Fgf8* and also downregulated was the distal *Hox* gene *a-13.* The upregulated Hox genes we identified were *Hoxb-6*, known to be upregulated by a pan-RAR agonist in the chick limb bud [34] and *Hoxd-3*. The latter Hox gene has been shown to increase the adhesiveness of human hematopoietic cell lines and induce the formation of cell aggregates by upregulating integrin a2/b3 [35]. Remarkably, integrins were the top canonical pathway by IPA seen here and, furthermore, the formation of cell aggregates is precisely the behavior that distal blastemal cells undergo during the proximalization process induced by RA [5]. These data indicate the proximalization of distal blastema tissue both in terms of its cellular behavior and positional control on the RARα pathway and imply a uniformity of patterning mechanisms across developing and regenerating vertebrate limbs.

Identifying genes related to the cell surface was the center of our analysis, not only because of this known change in adhesive behavior of blastemal cells, perhaps induced by *Hoxd-3*, but also because this is considered to be the mechanism by which positional information is assessed during regeneration (see Introduction). In support of this concept, as mentioned above, the top canonical pathway identified by IPA from our dataset was integrin signaling and six integrins genes were upregulated. Among the extracellular matrix-related genes, *Mmp21* was the second highest upregulated gene and the *collagen 5* and *6* genes were upregulated along with the downregulation of *collagen 4*. The cell membrane genes upregulated include *EphA5*, *Unc5a*, a netrin receptor, two cadherins *Cdh17* and *Cdh4* and genes for secreted proteins include neuron-derived neurotropic factor, a hyaluronan link protein, angiogenin, *Wnt4*, *Wnt6*, *Fgf2* and *Gdf6a*.

In relation to other genes considered to be responsible for the assessment of PI, we surprisingly did not find any differential expression of *Prod1*, the GPI-linked cell surface molecule identified in the newt, *Notophthalmus* [6]. This may be because in the axolotl, the homolog to newt *Prod1* has multiple stop codons generating two protein products with truncated C-terminals, both of which lack a GPI-anchor and are secreted instead of being localized to the cell surface [8]. Perhaps axolotl *Prod1* is not involved in positionally related events; rather, it may be involved in proliferation as a binding partner to nAG [36].

A more recent experiment in axolotls using a single cell sequencing dataset of connective tissue cells has identified a gene called *Rarres1* (retinoic acid receptor responder 1) or *Tig1* [37]. This gene is expressed in a shallow gradient in the mature limb, is upregulated by RA and is localized to the cell surface, having a transmembrane domain and a hyaluronic acid binding motif characteristic of its human homologue. Transfection of *Tig1* into the blastema induces the translocation of distal cells to proximal regions and an antibody to TIG1 prevents the engulfment of distal blastemas by proximal blastemas, precisely the behavior required of a specifier of positional information. We see the upregulation by 1.84-fold of *Tig1* in our dataset, suggesting that this gene is a downstream target of RARα which acts to regulate PD identity. Since this was a relatively low level of upregulation (and below our 2-fold cut-off), it suggests that searching for genes on the basis of their high levels of upregulation may not the most successful strategy.

Much work will be required to test the function of the individual genes we have identified above in controlling positional information, their pathways from the genome to the cell surface and ECM and how these molecules interact, but our results described here provide continuing supporting for the concept that in regeneration, the cell surface is the seat of positional information recognition.

## 5. Conclusions

Our data suggest that the RARα transcription factor is the dominant receptor which transduces the endogenous RA signal to control PD identity in the regenerating limb. The complex transcriptional network, which is instigated in respecifying PI, involves many components: (1) the upregulation of retinoic acid metabolism processes such as the Cyp26s to autoregulate the RA signal; (2) the downregulation of cell cycling; (3) the downregulation of distal positional genes such as *Hoxa*-*13* and of distal regulatory secreted molecules such as Fgf-8; (4) the upregulation of proximal identity genes such as *Meis1* and *2*; (5) the upregulation of ECM regulators such as Mmp21 and ECM molecules such as collagen 5 and 6; (6) the upregulation of cell surface molecules such as integrins, cadherins, ephrins; (7) the release of secreted molecules such as Wnts, neurotrophic factors and angiogenic factors. How these pathways interact will be a major task for continuing research on the nature and control of positional information in the regenerating limb.

## Figures and Tables

**Figure 1 cells-10-02174-f001:**
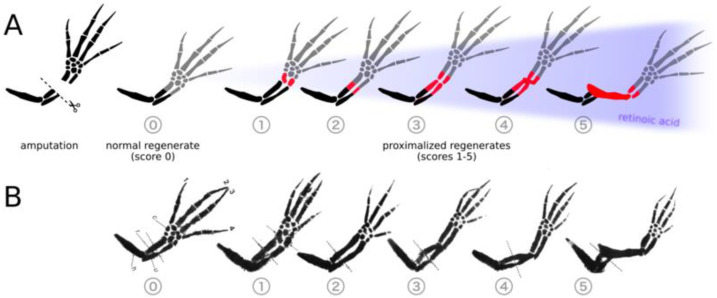
Retinoic acid induces the proximalization of regenerating tissue and duplicates limb elements along the PD axis. (**A**) Following amputation of the axolotl forelimb at the mid-radius/ulna, the regenerate (gray) is identical to the tissue lost. If the regenerating limb is treated with retinoic acid, more tissue is regenerated and limb elements are duplicated (red). Regenerates 1 through 5 depict five possibilities, each showing higher levels of duplication caused by increasing concentrations of RA (blue gradient, increasing from left to right). Regenerate 5 shows complete duplication of the humerus, a result indicative of proximalization to the shoulder level; (**B**) Images of reference limb regenerates scored from normal (score 0) to complete limb duplication (score 5), reproduced from [5].

**Figure 2 cells-10-02174-f002:**
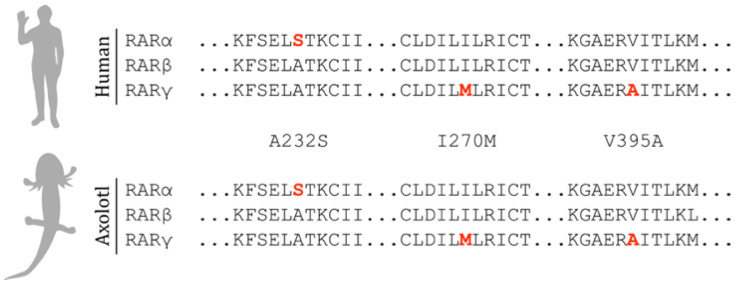
The three retinoic acid receptors (RARα, RARβ, and RARγ) are homologous, as seen in the alignments of select regions of the ligand-binding domain of both human (top) and axolotl (bottom). Three amino acid substitutions, depicted in red, grant binding specificity to the RAR agonists Am580, CD2019, and CD437. The protein sequences were obtained from the EST database hosted at Sal-Site [26].

**Figure 3 cells-10-02174-f003:**
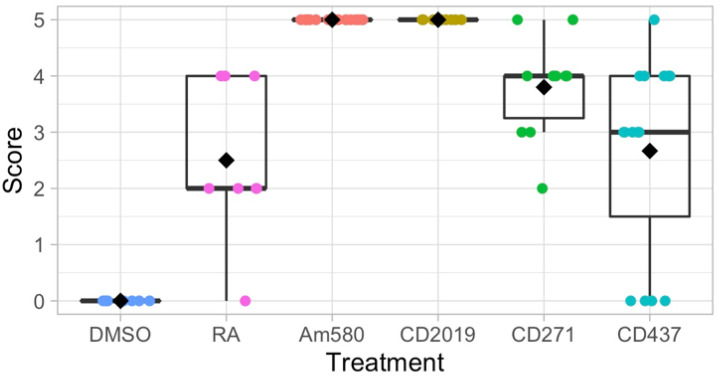
Duplication of forelimb regenerates from juvenile axolotls (3–5 cm) treated systemically with 250 nM Am580 (*n* = 20 limbs), CD2019 (*n* = 10), CD271 (*n* = 10), or CD437 (*n* = 15). Retinoic acid (*n* = 8) and DMSO (*n* = 8) served as the positive and negative controls, respectively. A score of 0–5 was assigned to each limb based on the degree of duplication (Figure 1). Limbs treated with Am580 and CD2019 were all completely duplicated, while CD271 and CD437 induced a range of duplication. All treatments have greater duplication than the DMSO control (DMSO vs. RA, *p* = 6.08 × 10^−4^; DMSO vs. CD271, *p* = 9.1 × 10^−6^; DMSO vs. CD437, *p* = 4.5 × 10^−4^). Am580 and CD2019 are greater than RA (RA vs. Am580 or CD2019, *p* = 1.9 × 10^−3^; RA vs. CD271). CD271 is just significantly more effective than RA (*p* = 0.016) and CD437 is not significantly more effective than RA (*p* = 0.438).

**Figure 4 cells-10-02174-f004:**
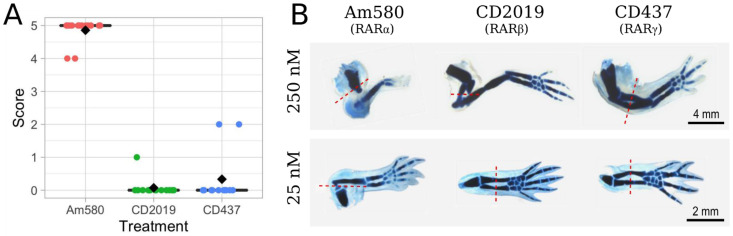
Comparison of PD limb duplication at 250 nM and 25 nM systemic agonist treatment. (**A**) Scoring following Victoria blue staining of mid-zeugopod limb regenerates treated with 25 nM agonist reveals a decrease in duplication to nearly zero in both CD2019 (*n* = 14 limbs; *p* = 9.4 × 10^−10^) and CD437 (*n* = 12; *p* = 5.1 × 10^−4^). There is no significant difference between Am580 (*n* = 14) treatments at 250 nM (Figure 3) and 25 nM (*p* = 0.082); (**B**) Representative limb regenerates following systemic exposure to 250 nM (upper panel) or 25 nM RAR (lower panel) agonists. The amputation plane is shown in red. The photos were processed by removing the background and adjusting the brightness.

**Figure 5 cells-10-02174-f005:**
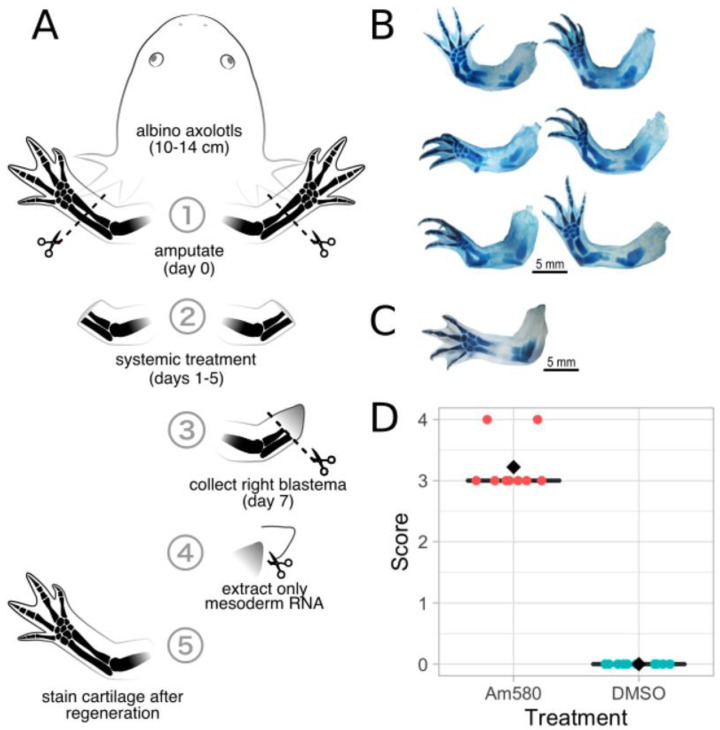
(**A**) Experimental design for both DMSO control (*n* = 10) and 25 nM RARα (*n* = 10) treatments. Both forelimbs are amputated mid-zeugopod at day 0 and treated systemically with either DMSO or Am580 on days 1–5. The right blastema is collected on day 7, the epidermis removed, and RNA extracted from the mesoderm. The left blastema serves as an observational control. (**B**) Left forelimb regenerates collected from the six axolotls that were selected for RNA-sequencing; (**C**) A representative left forelimb regenerate collected from a DMSO-treated axolotl. (**D**) Limb duplication scores for all left limbs in the Am580-treatment and DMSO-control groups. The Am580-treated limbs scored 3.2, an increase in PD duplication compared to DMSO controls (mean score of 0).

**Table 1 cells-10-02174-t001:** Top 30 differentially expressed transcripts. Transcript name is given as “AMEXTC” if found in the reference genome or “MSTRG” if identified as novel by StringTie. UniProt identification and gene names are given for transcripts with a significant BLASTx result.

Transcript Name	UniProt ID	Gene Name	log2FC
AMEXTC_0340000164913_cytochrome	CP26A_CHICK	CP26A	15.382878
AMEXTC_0340000004000_LOC108712603	MMP21_CYNPY	MMP21	8.68081722
AMEXTC_0340000018651_retinoic	RARB_HUMAN	RARB	8.34534428
MSTRG.18535	RS25_SHEEP	RPS25	6.96185493
AMEXTC_0340000220729_hypothetical	APMAP_RAT	APMAP	6.31133061
MSTRG.12045	ATTY_MOUSE	TAT	6.13262763
MSTRG.12046	ATTY_MOUSE	TAT	5.72243555
MSTRG.2544	-	-	5.61165445
AMEXTC_0340000053364_LOXe	LX15B_HUMAN	ALOX15B	5.26551299
AMEXTC_0340000039558_TAT	ATTY_MOUSE	TAT	5.26351744
mRNA01059	-	-	5.21247926
AMEXTC_0340000032012_LOC108645441	-	-	5.15846088
AMEXTC_0340000247464_ripply3.L	DSCR6_XENLA	RIPPLY3	5.05107637
MSTRG.17898	NDNF_HUMAN	NDNF	5.04336304
AMEXTC_0340000179221_LOC102944588	GIMA4_HUMAN	GIMAP4	5.02378211
AMEXTC_0340000012549_fibroblast	FGF8_CHICK	FGF8	−4.8925275
MSTRG.4054	-	-	−4.9088476
MSTRG.23812	-	-	−4.9178374
MSTRG.16115	-	-	−4.966291
MSTRG.23482	RTJK_DROME	POL	−5.0176166
MSTRG.4440	TPH_XENLA	TPH	−5.2716375
AMEXTC_0340000028898_LOC108413561	RTJK_DROME	POL	−5.2869995
AMEXTC_0340000081353_OCSTAMP	OCSTP_HUMAN	OCSTAMP	−5.3164526
MSTRG.1117	-	-	−5.5152443
MSTRG.16116	LIN1_NYCCO	LIN1	−5.6941371
MSTRG.13846	-	-	−5.702535
AMEXTC_0340000024275_LOC101872464	B3A2_PONAB	SLC4A2	−5.8790351
AMEXTC_0340000007120_HTR6	5HT6R_PANTR	HTR6	−6.3188017
MSTRG.5087	-	-	−6.5656262
MSTRG.25599	LORF2_HUMAN	LIN1	−6.8496992

**Table 2 cells-10-02174-t002:** Integrin signaling genes. Identified as a top canonical pathway by IPA.

Gene Symbol	Entrez Gene Name	log2 Ratio
CAV1	caveolin 1	0.642
ITGA2	integrin subunit alpha 2	1.139
ITGA4	integrin subunit alpha 4	1.139
ITGA5	integrin subunit alpha 5	0.492
ITGA6	integrin subunit alpha 6	0.608
ITGA2B	integrin subunit alpha 2b	1.001
ITGAV	integrin subunit alpha V	−1.224
ITGAX	integrin subunit alpha X	1.723
ITGB2	integrin subunit beta 2	−0.904
ITGB3	integrin subunit beta 3	−1.307

**Table 3 cells-10-02174-t003:** Genes encoding membrane-bound proteins. The transcript name is given as “AMEXTC” if the transcript is found in the reference genome or “MSTRG” if identified as novel by StringTie. The UniProt ID and gene name are given for the most similar homolog as determined by BLASTx search. The log2 fold change was determined by edgeR, with positive values indicating upregulated expression following Am580 treatment.

Transcript	UniProt ID	Gene Name	log2 FC
AMEXTC_0340000038644_LOC108803933	P2RY1_MOUSE	P2RY1	4.48
AMEXTC_0340000062820_GEM	GEM_HUMAN	GEM	4.32
MSTRG.21040	HRH2_MOUSE	HRH2	3.32
MSTRG.22927	TLR4_PIG	TLR4	3.09
AMEXTC_0340000034284_LOC102448090	M4A4A_HUMAN	MS4A4A	3.05
AMEXTC_0340000013374_htr2b	5HT2B_HUMAN	HTR2B	2.93
AMEXTC_0340000062083_FGD2	FGD2_MOUSE	FGD2	2.83
MSTRG.8675	FGD2_HUMAN	FGD2	2.71
MSTRG.3167	FSHR_CAIMO	FSHR	2.47
AMEXTC_0340000014713_EPHA5	EPHA5_HUMAN	EPHA5	2.4
AMEXTC_0340000042498_ADRA2A	ADA2A_PIG	ADRA2A	2.3
AMEXTC_0340000030234_LOC109141183	UNC5A_HUMAN	UNC5A	2.24
AMEXTC_0340000057843_LOC108718813	MRC1_HUMAN	MRC1	2.2
AMEXTC_0340000156303_BLNK	BLNK_CHICK	BLNK	2.12
AMEXTC_0340000062611_RGR	RGR_BOVIN	RGR	2.06
AMEXTC_0340000150384_cadherin	CAD17_MOUSE	CDH17	3.28
AMEXTC_0340000048099_CDH4	CADH4_CHICK	CDH4	3.11
AMEXTC_0340000004184_LOC102348750	CEAM5_MOUSE	CEACAM5	2.66
AMEXTC_0340000028915_ICAM5	ICAM5_MOUSE	ICAM5	2.44
AMEXTC_0340000049914_Sell	LYAM2_HUMAN	SELE	2.39
AMEXTC_0340000220626_PLLP	PLLP_HUMAN	PLLP	2.3
AMEXTC_0340000220764_LOC101952950	MYO10_BOVIN	MYO10	2.06

**Table 4 cells-10-02174-t004:** Genes encoding secretory proteins. Transcript name is given as “AMEXTC” if found in the reference genome or “MSTRG” if identified as novel by StringTie. Gene names are left blank for UniProt entries without a listed gene name.

Transcript	UniProt ID	Gene Name	log2FC
AMEXTC_0340000004000_LOC108712603	MMP21_CYNPY	MMP21	8.68
MSTRG.17898	NDNF_HUMAN	NDNF	5.04
MSTRG.6144	FCNV4_CERRY	-	4.36
AMEXTC_0340000030622_PROZ	PROZ_BOVIN	PROZ	4.22
AMEXTC_0340000250114_c4a.L	CO4_BOVIN	C4	3.87
AMEXTC_0340000209130_LOC105403673	CO5A1_MOUSE	COL5A1	3.63
AMEXTC_0340000025257_HAPLN4	HPLN4_MOUSE	HAPLN4	3.55
MSTRG.15067	VCO3_NAJKA	-	3.42
AMEXTC_0340000036400_LOC100552635	FCNV1_VARKO	FCNV1_VARKO	3.41
AMEXTC_0340000044633_PKDCC	PKDCC_HUMAN	PKDCC	3.29
AMEXTC_0340000062173_Angiogenin	ANGI_MOUSE	ANG	3.22
MSTRG.6402	MSMB_PIG	MSMB	3.17
AMEXTC_0340000056401_ostn	OSTN_HUMAN	OSTN	3.09
MSTRG.829	IL12B_FELCA	IL12B	2.85
AMEXTC_0340000250115_complement	CO4_RAT	C4	2.6
AMEXTC_0340000233392_CETP	CETP_CHICK	CETP	2.6
AMEXTC_0340000044847_Coiled-coil	WNT6_MOUSE	WNT6	2.48
AMEXTC_0340000048200_TSPEAR	TSEAR_HUMAN	TSPEAR	2.47
MSTRG.13661	IL18_CHICK	IL18	2.36
MSTRG.10022	IL8_CHICK	IL8	2.32
AMEXTC_0340000000382_fibroblast	FGF2_BOVIN	FGF2	2.14
MSTRG.1555	GDF6A_DANRE	GDF6A	2.14
AMEXTC_0340000053380_COL27A1	CO5A3_HUMAN	COL5A3	2.09
AMEXTC_0340000232220_LOC107293967	CHIT1_HUMAN	CHIT1	2.07
AMEXTC_0340000170356_LOC102365657	SFTPD_RAT	SFTPD	2.05
AMEXTC_0340000064262_ndnf	NDNF_XENTR	NDNF	2.04
MSTRG.5658	SDF1_XENLA	SDF1	2.03
AMEXTC_0340000010931_ADAMTS3	ATS3_HUMAN	ADAMTS3	2.03
MSTRG.22754	HMCN1_HUMAN	HMCN1	2.01

## Data Availability

The data is now publicly available and has been assigned accession GSE182296 at https://www.ncbi.nlm.nih.gov/geo/query/acc.cgi?acc=GSE182296 (accessed on 19 August 2021).

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
