# Peer review of "Retinoic Acid Receptors and the Control of Positional Information in the Regenerating Axolotl Limb"

_cells, 2021, doi:10.3390/cells10092174_

Round 1

Reviewer 1 Report

The paper by Polvadore and Maden titled “Retinoic acid receptors and the control of positional information in the regenerating axolotl limb” aims to uncover genes responsible for the proximodistal identity of blastema cells during axolotl limb regeneration. The authors used retinoic acid, a molecule known to induce proximal limb structures, agonists that signal through specific receptors. Using a decreasing concentration of these agonists, they aimed to reduce background effects and provide a strong (or stronger/more relevant than previous studies) dataset of genes that respond to RA signaling.

This study provides a resource of candidate genes that respond to RA signaling through the RARa receptor resulting in proximalization of the regenerating limb. Secondarily, it has demonstrated the effects of various RA agonists in limb proximalization. I personally believe that this is important information for the axolotl limb regeneration field. However, I am troubled by the inconsistencies on animal length used throughout the paper: Figure 3 uses 3-5cm, Figure 4 uses 6-8cm and the RNAseq uses 10cm. In addition, the study does not adequately meet one of its aims: to provide a more comprehensive list of candidate genes versus previous studies. Is this dataset important: Yes, but there are no real data supporting that the dataset is more useful than the ones previously generated --- Probably they are all equally important. In all cases what is lacking is functional studies. These cannot be addressed within the scope of the current paper without significant experimentation.

The following minor changes would make the paper better:

  • Figure 1 shows duplication scores and analysis using cartoon limbs. Authors should provide an actual representative limb photo for each score and specific skeletal features that were used to determine this score. For instance, scores 1-2 and 3-4 are not very clear how they are scored.
  • From the data analysis in Figure 3 and 4 it is not clear whether one animal had two amputations and generated two data points or whether these were averaged so one animal provided one data point. This is important since Figure 3 has binary scores of 0 – 1 – 2 – 3 – 4 – 5 but in Figure 4 one of the data points is at 2.5.
  • There are no statistics for Figure 3 and Figure 4.
  • There are no scale bars in stained limb structure photos.
  • Figure 4 shows photos from two different experiments using different size animals. This should be noted in the legend more specifically.
  • Figure 5 do not show DMSO control limbs.

Author Response

Thank you for the valuable comments on our ms.

This study provides a resource of candidate genes that respond to RA signaling through the RARa receptor resulting in proximalization of the regenerating limb. Secondarily, it has demonstrated the effects of various RA agonists in limb proximalization. I personally believe that this is important information for the axolotl limb regeneration field. However, I am troubled by the inconsistencies on animal length used throughout the paper: Figure 3 uses 3-5cm, Figure 4 uses 6-8cm and the RNAseq uses 10cm. In addition, the study does not adequately meet one of its aims: to provide a more comprehensive list of candidate genes versus previous studies. Is this dataset important: Yes, but there are no real data supporting that the dataset is more useful than the ones previously generated --- Probably they are all equally important. In all cases what is lacking is functional studies. These cannot be addressed within the scope of the current paper without significant experimentation.

The following minor changes would make the paper better:

  • Figure 1 shows duplication scores and analysis using cartoon limbs. Authors should provide an actual representative limb photo for each score and specific skeletal features that were used to determine this score. For instance, scores 1-2 and 3-4 are not very clear how they are scored.

An additional figure (1B) with real limbs is shown for additional clarity as requested.

  • From the data analysis in Figure 3 and 4 it is not clear whether one animal had two amputations and generated two data points or whether these were averaged so one animal provided one data point. This is important since Figure 3 has binary scores of 0 – 1 – 2 – 3 – 4 – 5 but in Figure 4 one of the data points is at 2.5.

Thank you for this. We have gone back to look at the raw data and seen that these 3 points in the RA data were wrongly scored. This has now been corrected.

  • There are no statistics for Figure 3 and Figure 4.

These have now been added.

  • There are no scale bars in stained limb structure photos.

Scale bars have now been added to these figs.

  • Figure 4 shows photos from two different experiments using different size animals. This should be noted in the legend more specifically.

This has been added to the legend.

  • Figure 5 do not show DMSO control limbs.

A DMSO control limb has now been added.

Reviewer 2 Report

In regenerating limbs of the axolotl the blastema produces new cells with positional identity corresponding to more distal identities than the level of amputation. However, when blastemas are treated with retinoic acid (RA), cells of more proximal fates are initially produced, a process also known as super-regeneration. The molecular mechanisms underlying this behavior have been elucidated in newt, but the homolog of the RA-regulated prod1 gene in axolotl does not have the same direct effects on the blastema. The authors of the present manuscript sought to identify genes that are regulated by RA in axolotl.

In the first part, they characterized the ability of RAR-subtype specific agonists to change positional information in the blastema and could show that both RARa- and RARb-specific agonists are sufficient to induce the desired effects, but that only the RARalpha-specific agonist does so at a concentration reduced by an order of magnitude.

This concentration was used to elicit shifts in positional identity in limb blastemas, extract RNA from the blastema mesenchyme and prepare cDNA for RNAseq. Following comparison with untreated blastemas, the authors identified more than 3000 regulated genes, of which a subset of 505 genes was characterized by an abundance of genes with presumed roles related to the cell surface. The identified genes provide a source for further studies aimed at identifying those genes that control positional information during axolotl limb regeneration.

The work represents a considerable advance in the bigger effort to identify the long sought-after molecules that confer positional identity in the limb, especially since the process in newts relies on a different set of molecules. Importantly, signaling through RARa is established as being sufficient to elicit proximalizations. The tables of regulated genes contains several ones with expected trends of regulation, as summarized in the conclusions, suggesting that genes of interest are likely to be contained in the dataset. Knowing that positional information in axolotl requires signals from the surface of mesenchymal cells, the authors identify extensive subsets of genes that qualify as regulators of surface properties. As a note of caution I would point out that Tig1/Rarres1 falls below the artificial log2 fold change threshold used for the top 30 candidates in this study. It therefore remains to be shown if “high regulation” is a necessary qualifier for PI genes.

Major issue:

  1. Please cite data that confirm that all 3 agonists elicit comparable activitation levels at the same concentration. If Am580 is more active at lower concentrations compared to the others, this does not necessarily mean that an essential part of PI is mainly controlled by RARa and it would remain possible that CD2019 needs a higher concentration to activate RARb.

Minor issues:

line 11: Rather than (spelling)

line 46: please specify more clearly around which axis the 180° rotation was performed.

line 111-114: please confirm that both forelimbs were used to extract RNA in experimental animals

line 225: EST database (spelling)

line 228: show (delete)

Figure 3: please provide the numbers of limbs analysed and confirm that dots represent data points from individual limbs.

line 454: Crabp2 were (spelling and italics)

line 462: in keeping (spelling)

Author Response

Thank you for the valuable comments on our ms.

In regenerating limbs of the axolotl the blastema produces new cells with positional identity corresponding to more distal identities than the level of amputation. However, when blastemas are treated with retinoic acid (RA), cells of more proximal fates are initially produced, a process also known as super-regeneration. The molecular mechanisms underlying this behavior have been elucidated in newt, but the homolog of the RA-regulated prod1 gene in axolotl does not have the same direct effects on the blastema. The authors of the present manuscript sought to identify genes that are regulated by RA in axolotl.

In the first part, they characterized the ability of RAR-subtype specific agonists to change positional information in the blastema and could show that both RARa- and RARb-specific agonists are sufficient to induce the desired effects, but that only the RARalpha-specific agonist does so at a concentration reduced by an order of magnitude.

This concentration was used to elicit shifts in positional identity in limb blastemas, extract RNA from the blastema mesenchyme and prepare cDNA for RNAseq. Following comparison with untreated blastemas, the authors identified more than 3000 regulated genes, of which a subset of 505 genes was characterized by an abundance of genes with presumed roles related to the cell surface. The identified genes provide a source for further studies aimed at identifying those genes that control positional information during axolotl limb regeneration.

The work represents a considerable advance in the bigger effort to identify the long sought-after molecules that confer positional identity in the limb, especially since the process in newts relies on a different set of molecules. Importantly, signaling through RARa is established as being sufficient to elicit proximalizations. The tables of regulated genes contains several ones with expected trends of regulation, as summarized in the conclusions, suggesting that genes of interest are likely to be contained in the dataset. Knowing that positional information in axolotl requires signals from the surface of mesenchymal cells, the authors identify extensive subsets of genes that qualify as regulators of surface properties. As a note of caution I would point out that Tig1/Rarres1 falls below the artificial log2 fold change threshold used for the top 30 candidates in this study. It therefore remains to be shown if “high regulation” is a necessary qualifier for PI genes.

The latter point is a very valuable one and we have added words to this effect in lines 516-518

Major issue:

  1. Please cite data that confirm that all 3 agonists elicit comparable activitation levels at the same concentration. If Am580 is more active at lower concentrations compared to the others, this does not necessarily mean that an essential part of PI is mainly controlled by RARa and it would remain possible that CD2019 needs a higher concentration to activate RARb.

 This has been added in line 269 and ref [11] contains this data.

Minor issues:

line 11: Rather than (spelling)

Corrected

line 46: please specify more clearly around which axis the 180° rotation was performed.

‘around the circumferential axis’ has been added to clarify

line 111-114: please confirm that both forelimbs were used to extract RNA in experimental animals

The design of this experiment has been clarified in section 2.4.1. In these experimental animals the right limb blastema was used for sequencing (mesenchyme only) and the left limb of each animal was allowed to complete regeneration to ensure that the respecification had occurred (in Am580 treated limbs) or were normal (in DMSO treated limbs). This was designed specifically to ensure that appropriate respecification had indeed occurred in this experiment because contralateral limbs on any one animal always behave identically in our experience.

line 225: EST database (spelling)

Corrected

line 228: show (delete)

Corrected

Figure 3: please provide the numbers of limbs analysed and confirm that dots represent data points from individual limbs.

Individual n numbers are now recorded in the legend and vary from n=8 to n=20. Dots do represent individual data points but when they have the same value they merge together thus making them impossible to identify individually.

line 454: Crabp2 were (spelling and italics)

Corrected

line 462: in keeping (spelling)